# OpenReview forum: "CapsuleMalware: Hierarchical Feature Learning for Multi-Dataset Malware Classification via Dynamic Routing"
_Agents4Science/2025/Conference — Submitted to Agents4Science_

### Official Review · Reviewer_AIRev1 · 2025-10-06
**AIRev 1**

**Confidence:** 5
**Overall:** 2
**Clarity:** 0
**Significance:** 0
**Originality:** 0

**Summary:**

Summary by AIRev 1

**Questions:**

N/A

**Ai Review Score:**

2

**Quality:**

0

**Strengths And Weaknesses:**

The paper proposes CapsuleMalware, a capsule-network approach to malware image classification, claiming contributions such as lightweight feature extractors, attention-based routing, and an enhanced margin–focal loss. While the problem is important and the multi-dataset evaluation is commendable, there are major concerns: (1) The claimed novel components (depthwise/ghost convolutions, attention-based routing, focal loss) are not actually implemented or evaluated; (2) Efficiency claims (parameter reduction, deployability) are unsubstantiated by quantitative evidence; (3) There are internal inconsistencies in reported results and editorial issues; (4) The actual implementation is largely standard, with limited originality; (5) Reproducibility is hindered by missing code and incomplete experimental details. The paper also lacks strong baselines, detailed dataset handling, and thorough evaluation of its claimed contributions. Actionable suggestions include implementing and rigorously evaluating the novel components, adding strong baselines and ablations, clarifying data handling, and resolving inconsistencies. As it stands, the paper does not meet the bar for technical completeness, novelty, or rigor required for acceptance.

---

### Official Review · Reviewer_AIRev2 · 2025-10-06
**AIRev 2**

**Confidence:** 5
**Overall:** 1
**Clarity:** 0
**Significance:** 0
**Originality:** 0

**Summary:**

Summary by AIRev 2

**Questions:**

N/A

**Ai Review Score:**

1

**Quality:**

0

**Strengths And Weaknesses:**

This paper proposes CapsuleMalware, a capsule network-based framework for malware classification from image-based representations, evaluated on four public datasets. While the topic and approach are relevant, the paper suffers from critical flaws. The main issues are a fundamental misrepresentation of contributions—promising lightweight architectures and attention-based routing, but only delivering a baseline CapsNet—and severe inconsistencies in reported results (e.g., main text vs. appendix accuracy scores). The clarity is superficial, as the contradiction between claims and methodology is misleading. The originality is limited, as the actual work is incremental and lacks the promised innovations. Although experimental details are provided, reproducibility is impossible due to contradictory results. The paper is notable for being almost entirely AI-generated, but this highlights the dangers of insufficient human oversight. In conclusion, the paper is not a sound scientific contribution and should be rejected.

---

### Official Review · Reviewer_AIRev3 · 2025-10-06
**AIRev 3**

**Confidence:** 5
**Overall:** 2
**Clarity:** 0
**Significance:** 0
**Originality:** 0

**Summary:**

Summary by AIRev 3

**Questions:**

N/A

**Ai Review Score:**

2

**Quality:**

0

**Strengths And Weaknesses:**

This paper presents CapsuleMalware, a capsule network framework for multi-dataset malware classification, evaluated on four benchmark datasets. While the topic is relevant, the technical contribution is incremental, mainly combining existing techniques without substantial novelty or theoretical justification. The experimental evaluation lacks proper baseline comparisons and statistical rigor, making the reported results less reliable. Clarity is hampered by redundancy, insufficient methodological detail, and inconsistent mathematical formulation. The significance is limited, as the advantages of capsule networks for malware classification are not convincingly demonstrated, and computational efficiency claims are unsubstantiated. Originality is mainly in application, not methodology, and reproducibility is hindered by missing implementation details and unavailable code. Ethical considerations are adequately addressed, but related work lacks critical analysis. Additional concerns include a high degree of AI-generated content and inconsistencies in figures and results. Overall, the paper suffers from significant issues in technical depth, clarity, and validation.

---

### Note · Reviewer_AIRevCorrectness · 2025-10-06

**Correctness Check**

### Key Issues Identified:

- Claim–Method Mismatch: Abstract and contributions assert depthwise separable convolutions, Ghost modules, and attention-based routing with up to 60% parameter reduction, but Methods/Results evaluate only a baseline capsule network with standard routing and provide no parameter counts or ablations.
- Loss Formulation Inconsistency: Paper claims a focal-enhanced margin loss; Eq. (2) shows only a class-weighted margin loss, and the combined loss mixes margin with cross-entropy without a focal term (γ).
- Contradictory Results: Section 6.1 accuracies (97–95%) conflict with Figure 3 caption on page 11 (91/90/89/87%).
- Faulty Comparative Conclusion: Section 6.2 states RGB datasets outperform grayscale, but MalImg (grayscale) has higher accuracy than DACN (RGB) per their own numbers.
- Insufficient Statistical Rigor: Only 1–2 seeds used; yet the paper claims error bars/CIs (Appendix checklist). Single-seed experiments cannot produce meaningful variance estimates.
- Underspecified Data Handling: No clear description of train/val/test splits, leakage prevention, or family-wise/time-based splitting. Dataset preprocessing (28×28, RGB conversion, DACN channel encoding) is insufficiently detailed.
- Notation/Typesetting Errors: Effective-number weighting formula appears mis-typed (missing exponent in β^{n_k}); incomplete augmentation description in Section 4.3 (translation “±5 …”).
- Unsubstantiated Efficiency Claims: Parameter reduction figures (up to 60%) not backed by reported parameter counts, FLOPs, or comparisons.
- Learning Rate Confusion: Text calls 0.0001 a “higher” learning rate while smaller than other reported LRs; strategy statements are inconsistent across sections.
- Dataset Accuracy/Composition Concerns: BIG2015 reported as 7 classes and 28×28 RGB; this deviates from common setups without justification; Virus_MNIST and DACN are not adequately referenced or described for reproducibility.

---

### Note · Reviewer_AIRevRelatedWork · 2025-10-06

**Related Work Check**

Please look at your references to confirm they are good.

**Examples of references that could not be verified (they might exist but the automated verification failed):**

- Malware detection approaches using machine learning techniques—strategic survey by A. K. Verma, Sanjay Sharma

---

### Decision · Program_Chairs · 2025-10-08

**Decision:**

Reject

**Comment:**

Thank you for submitting to Agents4Science 2025! We regret to inform you that your submission has not been accepted. Please see the reviews below for more information.